# On the sensitivity of the Devonian climate to continental configuration, vegetation cover and insolation

Julia Brugger<sup>1,2,3</sup>, Matthias Hofmann<sup>1</sup>, Stefan Petri<sup>1</sup>, and Georg Feulner<sup>1</sup>

<sup>1</sup>Potsdam-Institut für Klimafolgenforschung (PIK), Mitglied der Leibniz-Gemeinschaft, Postfach 60 12 03, 14412 Potsdam, Germany

<sup>2</sup>Universität Potsdam, Institut für Physik und Astronomie, Karl-Liebknecht-Straße 24/25, 14476 Potsdam, Germany <sup>3</sup>Berlin-Brandenburgisches Institut für Biodiversitätsforschung, Altensteinstraße 34, 14195 Berlin, Germany

Correspondence to: Julia Brugger (brugger@pik-potsdam.de) and Georg Feulner (feulner@pik-potsdam.de)

**Abstract.** During the Devonian period (419 to 359 million years ago), life on Earth witnessed decisive evolutionary breakthroughs, most prominently the colonisation of land by vascular plants and vertebrates. At the same time, it is also a period of major marine extinction events coinciding with marked changes in climate. There is limited knowledge about the causes of these changes and their interactions. It is therefore instructive to explore systematically how the Devonian climate system

- responds to changes in critical boundary conditions. Here we use coupled climate-model simulations to investigate separately the influence of changes in orbital parameters, continental configuration and vegetation cover on the Devonian climate. Variations of Earth's orbital parameters affect the Devonian climate system, with the warmest climate states at high obliquity and high eccentricity, but the amplitude of global temperature differences is smaller than suggested by an earlier study based on an uncoupled atmosphere model. The prevailing mode of climate variability on decadal to centennial timescales relates to sur-
- face air temperature fluctuations in high northern latitudes which are mediated by coupled oscillations involving sea-ice cover, ocean convection and a regional overturning circulation in the Arctic. Furthermore, we find only a small biogeophysical effect of changes in vegetation cover on global climate during the Devonian, and the impact of changes in continental configuration is small as well. Assuming decreasing atmospheric carbon dioxide concentrations throughout the Devonian, we then set up model runs representing the Early, Middle and Late Devonian. Comparing the simulations for these timeslices, the temperature
- evolution is dominated by the strong decrease in atmospheric carbon dioxide. In particular, the albedo change due to the increase in land vegetation alone cannot explain the temperature rise found in Late Devonian proxy data which remains difficult to reconcile with reconstructed falling carbon-dioxide levels. Simulated temperatures are significantly lower than estimates based on oxygen-isotope ratios, suggesting a lower  $\delta^{18}$ O ratio of Devonian seawater.

Copyright statement.

### 1 Introduction

The Devonian (419 to 359 Ma, 1 Ma = 1 million years ago) is a key period in Earth's history characterised by fundamental changes in the atmosphere, the ocean and the biosphere. With respect to the evolution of life, it is best known for the diversification of fish as well as the colonisation of the continents by vascular plants and vertebrates. Although microbial mats and nonvascular plants could be found on land even before the Devonian (Boyce and Lee, 2017), the appearance of vascular land plants beginning in the Early and Middle Devonian was certainly an important first step towards modern land ecosystems. By the Late Devonian, vascular plants had vastly diversified, going hand in hand with the evolution of more advanced leaves and root systems (Algeo et al., 1995; Algeo and Scheckler, 1998). In the ocean, coral stromatoporoid reefs reached their largest extent during the Phanerozoic in the Middle Devonian (Copper and Scotese, 2003) and fish evolved into rich diversity (Dahl et al., 2010). Finally, in the Late Devonian, the first tetrapods moved from ocean to land (Clack, 2007; Brezinski et al., 2009).

While the Devonian is best known for these evolutionary breakthroughs, it is also a period of species mass extinctions. The extinction rate during the Devonian is marked by three distinct peaks (Bambach, 2006), with the highest pulse ranking among the five most severe mass extinctions in Earth's history (Frasnian-Famennian mass extinction, 378–375 Ma, Bambach 2006). The cause of these extinctions, which mostly took place in the ocean (Bambach, 2006), is still under debate, as it

15 is challenging to explain the episodic nature and duration of the extinctions (Algeo et al., 1995). Discussed causes for the Frasnian-Famennian extinction are a bolide impact (McGhee et al., 1984), volcanic activity (Ma et al., 2016), changes in sea level (Bond and Wignall, 2008; Ma et al., 2016), rapid temperature variations (Ma et al., 2016) and the development of ocean anoxic waters (Bond and Wignall, 2008; Ma et al., 2016).

The multitude of remarkable biospheric changes during the Devonian occurred against a backdrop of considerable changes in atmospheric composition. In particular, carbon dioxide (CO<sub>2</sub>) concentrations decreased strongly from ~2,000 ppm to ~1,000 ppm (Foster et al., 2017) as shown in Fig. 1a. Note that this recent CO<sub>2</sub> compilation agrees with older compilations in the general decreasing trend during this time, but reports lower CO<sub>2</sub> concentrations for the Devonian than older proxy studies (Royer, 2006); they are also much lower than the CO<sub>2</sub> concentrations based on the GEOCARB model (Berner, 1994, 2006) frequently used in earlier modeling studies. In contrast to the decrease in carbon dioxide, oxygen levels witnessed an exceptionally strong rise about 400Ma (Dahl et al., 2010).

These changes in atmospheric composition, and in particular the drop in atmospheric  $CO_2$  concentrations, also resulted in climatic changes, which in turn affected the biosphere. Indeed,  $\delta^{18}O$  oxygen isotope data (shown in Fig. 1b) indicate a greenhouse climate in the Early Devonian and much cooler temperatures in the Middle Devonian (Joachimski et al., 2009). For the Late Devonian, proxy studies (van Geldern et al., 2006; Joachimski et al., 2009) indicate rising temperatures again which

30 are still challenging to reconcile with the decreasing CO<sub>2</sub> concentrations. During the late Famennian, Earth's climate cooled again (Brezinski et al., 2009; Joachimski et al., 2009), with some studies even indicating glaciations (Caputo, 1985; Caputo et al., 2008; Brezinski et al., 2008, 2009). Whether this is accompanied by a sea-level transgression linked to the development of ocean anoxic conditions (Johnson et al., 1985; Bond and Wignall, 2008) or a sea-level drop (Sandberg et al., 2002; Haq

**Figure 1.** (a) Atmospheric CO<sub>2</sub> concentration in the Devonian after Foster et al. (2017). The black circles represent the data from different proxies. The thick black line is the most likely LOESS fit taking into account the uncertainties in both age and CO<sub>2</sub>. The dark and light grey bands show the 68% and 95% confidence intervals. (b)  $\delta^{18}$ O values from conodont apatite from Joachimski et al. (2009), but using a different NBS120c standard for calibration (Lécuyer et al., 2003). Note the inverted scale on the *y* axis since an increase in  $\delta^{18}$ O translates in a temperature decrease. The light pink shading indicates the time intervals of the three Devonian periods of mass extinction: the late Givetian extinction from 389 to 385Ma, the Frasnian-Famennian extinction (or Kellwasser event) from 378 to 375Ma, and the Late Famennian extinction (Hangenberg event) from 364 to 359Ma (Bambach, 2006).

and Schutter, 2008; Ma et al., 2016) is still debated, but there is consent that the Late Devonian was a period of fast sea-level variations (Johnson et al., 1985; Sandberg et al., 2002; Haq and Schutter, 2008; Brezinski et al., 2009; Ma et al., 2016).

As any other period in Earth's history, the 60 million years spanning the Devonian are not only characterised by long-term changes in temperature, but also by large fluctuations in temperature around these longer-term trends (see Fig. 1b). Furthermore, there are several studies stressing the importance of astronomical forcing for the climate during the Devonian. The geologic record of the Devonian shows cyclic structures (De Vleeschouwer et al., 2013, 2017) which can be interpreted as the result of astronomical cycles according to Milankovitch theory (Milankovitch, 1941). The configuration of Earth's orbit and rotational axis determines the total amount as well as the spatial and temporal distribution of solar radiation, and therefore impacts

- axis determines the total amount as well as the spatial and temporal distribution of solar radiation, and therefore impacts climate. In addition, identifying astronomical cycles in the geologic record can help assigning a timescale to cyclic features observed in the geologic record, and thus a timescale for palaeoclimatic changes (De Vleeschouwer et al., 2013, 2014, 2017). In an effort to link the changes in the various components of the Devonian Earth system, there are several studies investigating
- potential connections between land plant evolution, climate change and the oceanic mass extinction (Berner, 1994; Algeo et al., 1995; Algeo and Scheckler, 1998; Godderis and Joachimski, 2004; Simon et al., 2007; Le Hir et al., 2011). It is suggested, for example, that the increase of weathering due to the spreading of plants on land could have been a cause of the decrease in carbon dioxide (Algeo et al., 1995; Algeo and Scheckler, 1998; Berner, 2006). Additionally, the increased weathering rates could have lead to a higher transport of phosphorus to the ocean, promoting eutrophication with its negative consequences for
- life in the ocean (Algeo et al., 1995; Algeo and Scheckler, 1998).

Given the multitude of changes in the Earth system during the Devonian and the intricate coupling of atmosphere, ocean and biosphere, it is challenging to disentangle causes and effects and to determine which forcings are most important for Devonian climate change. In this study, we test the sensitivity of the Devonian climate to different forcings in order to quantify their relevance. We therefore set up simulations with a coupled ocean–atmosphere model considering three different continental

configurations representing the Early, Middle and Late Devonian. In addition, changes in the solar constant, in atmo