# Peer review of "On the sensitivity of the Devonian climate to continental configuration, vegetation cover and insolation"

_Climate of the Past, 2018_

## Referee Comment (RC1) · Anonymous Referee #1 · 30 Apr 2018

**Recommendation**: Reject.

*Anonymous*

**Paper summary**:

Brugger et al. use the ocean-atmosphere model of intermediate complexity Climber-3α to quantify the sensitivity of the Devonian climate system to changes in the configuration of the continents, in vegetation cover and in orbital parameters. They also analyze patterns of quasi-periodic variations in global mean surface temperature arising from the waxing and waning of sea ice over the North Pole. In a 3rd step, they combine changes in the above-mentioned boundary conditions, plus $p$CO$_2$ and solar constant, to conduct 'best-guess' simulations, the purpose of which is to best capture the climatic conditions during the Early, Middle and Late Devonian (415, 380 and 360 Ma respectively). They focus their analysis on changes in surface air temperatures (SAT).

The authors demonstrate a weak impact of the paleogeographical evolution on globally-averaged SAT, although strong seasonal changes arise at the regional scale in response to the changing land-sea mask. Changes in vegetation cover, from bare land to coastal shrub and then trees, impact simulated mean annual SAT by no more than 0.2 °C. Over continents, the warming induced by the decrease in albedo is overwhelmed by the increase in evaporation and associated cooling effect. The main signal emerging from the sensitivity tests to the orbital configuration is the increase in global temperature with increasing obliquity and increasing eccentricity, whereas precession has a very minor impact. Regarding the analysis of the periodic flips between two climatic states over the North Pole, the authors suggest a mechanism linking sea-ice extent, ocean salinity and convective processes. The resulting climatic mechanism is associated with changes in global SAT on the order of 0.2 °C. Best-guess model runs feature a cooling trend throughout the Devonian, with mean annual globally-averaged SAT decreasing from 22.2 °C in the Early Devonian to 21.0 °C in the Middle Devonian and 19.3 °C in the Late Devonian. The authors explain that this trend is mainly due to the imposed drop in $p$CO$_2$ throughout the period (from 2000 ppmv to 1000 ppmv), while the impact of changes in vegetation cover can be neglected. These results contrast with a previous study that suggested that the climatic impact of decreasing $p$CO$_2$ throughout the Devonian may have been compensated by the concomitant decrease in albedo induced by the spread of land plants (Le Hir et al., 2011). Simulated tropical sea-surface temperatures are systematically lower than suggested by proxy data. Brugger et al. therefore suggest that the ocean $\delta^{18}$O composition may have been lower in the Devonian, thus biasing temperature reconstructions.

**Major comments**:

In this contribution, the authors conduct numerous sensitivity tests. They cover several topics previously investigated by other research groups using other climate models, namely the sensitivity to the vegetation cover and to the orbital configuration (Le Hir et al., 2011; De Vleeschouwer et al., 2014). As stated by the authors in their introduction, this is important to determine the response of the Devonian climate system to different kinds of perturbations, in order to better understand the mechanisms potentially lying behind the coupled changes in the ocean-atmosphere system and in the biosphere at that time. Such systematic experiments should also permit to determine to what extent the results obtained in previous studies are model-dependent. The manuscript is concise and well-written and the experimental setup, in spite of the numerous experiments and sensitivity tests, is well described so that the study is easy to follow. This is appreciated.

However, the paper does not convey any clear, robust message. The authors cover a wide spectrum of scientific questions and provide numerous experiments, but this generally goes with a lack of robust analysis of the model output. For this reason and other weaknesses exposed in detail

hereafter, I do not support publication of the manuscript in Climate of the Past. Below, I explain my motivations in rejecting this manuscript and I try to propose constructive comments.

**1**. On the choice of the numerical model. I do not like recommending using up-to-date, high-resolution models. Climate models of previous generations are useful because they generally provide results that are very close at first order to the ones obtained using more robust models while permitting numerous sensitivity tests due to their lower computational cost. This is obviously what motivated Brugger et al. to use Climber-3α. However, I am wondering if this ocean GCM coupled with a very coarse-resolution statistical-dynamical atmosphere model is really adapted to the kind of experiments the authors are conducting. Indeed, most of the signal analyzed by the authors is provided by the atmospheric component, be it in response to changing vegetation cover or orbital configuration. In that context, I would suggest adopting a relatively robust atmospheric model. I do not necessarily want to say that the authors should repeat their experiments with another model, but please, at least, clearly state the name of the model rather than hiding this behind a reference to Montoya et al. (2005) and include a discussion of the limitations. I also suggest clearly quantifying the bias associated with the interpolation onto the 22.5°x7.5° grid, see below.

**2**. On the weakness of the signal identified in the Arctic. The manuscript is generally well organized: the authors first investigate the impact of different forcing factors on simulated Devonian climate and subsequently combine these parameters to conduct their best-guess model runs. Section 3.4 'Flips between two climatic states in the Arctic' does not fit within this framework. It impedes the reading flow and, more critically, the amplitude of the SAT signal identified by the authors – ca. 0.2°C – is so weak that it would have no significant implications on the deep-time sedimentary record. I suggest deleting this part of the manuscript, which would at the same time allow the authors to expand on the other topics and thus to make their manuscript more robust in many aspects, see other major comments.

**3**. On the lack of sensitivity tests to the $p$CO$_2$. The rationale adopted by Brugger et al. more or less consists in decomposing the signal obtained in their best-guess model runs through individual sensitivity tests to the different parameters in play (continental configuration, vegetation cover…). In this context, it seems that a sensitivity test to the $p$CO$_2$ would be helpful. In particular, the authors state on page 18, lines 17-19 that "The global cooling over the Devonian seen in the set of best-guess simulations […] is dominantly driven by the strong decrease in CO$_2$ concentrations". This statement is not supported by the results provided in the current version of the manuscript. One could further imagine a series of best-guess experiments with $p$CO$_2$ held constant in order to clearly show that this constitutes the main driving factor of simulated climate cooling. The same could be done for the solar constant, the impact of which is not independently quantified. Also, the impact of changes in the orbital parameters is tested in Section 3.3 but it is not subsequently discussed. For instance, one could imagine showing the SST spread associated with different orbital configurations in Fig. 10.

**4**. On the need for a more robust analysis of the impact of the paleogeographical changes. In Section 3.1, the authors explain that changes in the land-sea mask induce strong seasonal SAT changes at the regional scale. What about the changes in land surface area and associated surface albedo? Please provide a table including key numbers such as land surface area for each timeslice. This is particularly important because results of the interpolation of the paleomaps shown in Fig. 2 onto the coarse grid of the atmospheric model are uncertain. Please check that the surface area on this grid is in first-order agreement with the original reconstructions provided by C. Scotese and plotted in Fig. 2.

**5**. On the need for a more robust analysis of the vegetation cover impact. The authors show that, in their simulations, the spread of Devonian land plants induces no critical changes in the mean annual,

globally-averaged SAT. These results are interesting because they contrast with previous work (Le Hir et al., 2011), which suggested that the decrease in albedo may have been sufficient to counter-balance the drop in $CO_2$. I think that this may constitute the most interesting part of the manuscript. However, several points need to be discussed/tested before drawing any robust conclusions:

- Unknown vegetation cover. Maps of imposed vegetation cover are not provided and the surface area occupied by these plants in the different experiments (i.e., coastal shrub and trees) is not specified. In other words, the boundary conditions are unknown. The authors describe how they built their maps in Section 2.2.2, but this remains obscure. For instance, they explain that they "assume 80 % grass-like vegetation cover in areas lower than 500 m in altitude and closer than 500 km to the coast". Well, what does it give on the 22.5°x7.5° grid? What's the magnitude of the bias (in terms of surface area) imposed by the interpolation onto this grid?

- Lack of justification for the definition of the plant spatial cover. Also, the authors do not justify their choices regarding the imposed land plant cover: why are trees absent in continental interiors? For instance, Le Hir et al. (2011) used a vegetation model and trees are simulated over the South Pole. Similarly, why is vegetation absent beyond 1500 m in altitude? This discussion is of first-order importance, since the choices made here drive an essential part of the model response to changing vegetation cover throughout the period.

- Lack of justification for the definition of the plant expansion scenario. Closely related to my previous point, I see no robust justification for the scenario imposed by the authors in their best-guess model runs. Le Hir et al. (2011) conducted an interesting review of the literature to propose a scenario of Devonian land plant evolution in their models. They identified several steps in the colonization of the land surface by Devonian plants and selected appropriate plant functional types accordingly for each time slice. The same kind of approach is required to ensure robust modeling results. I am particularly surprised that the authors impose a bare land in their Early Devonian best-guess model configuration: it is recognized that non-vascular plants appeared as soon as the Mid Ordovician (Rubinstein et al., New Phytologist 2010) and that vascular plants were already present by the Late Ordovician (Steemans et al., Science 2009), with possibly strong implications on the Earth System at that time (Lenton et al., NatGeo 2012). Modeling studies further suggest that non-vascular plant cover may have been relatively widespread during the Ordovician (Porada et al., NatCom 2016; Lenton et al., PNAS 2016). I encourage the authors to explore the literature to build their experimental design more robustly.

- Poorly documented plant properties. The properties of the vegetation cover are poorly documented. What are the values of evapotranspiration and roughness associated with each plant functional type in use? Changes in evapotranspiration are a primary driver of the simulated SAT changes. Values have to be discussed.

- Lack of justification for the plant properties. Also, do the plant properties correctly represent Devonian plants? The authors should discuss this and use their numerical model to investigate the range of possible values, in order to determine if their results are robust. This is a crucial point.

**Minor comments**:

**1**. Title of the manuscript: I suggest replacing 'insolation' with 'orbital configuration' to make it clear that this does not refer to the changes in solar constant, which are also accounted for in the best-guess model runs.

**2**. Page 4, lines 20-21: Changes in solar constant are not systematically investigated.

**3**. Please avoid throwaway sentences, e.g. on page 9, lines 12-13: "Furthermore, shifts in the distribution of land and ocean areas as well as changes in ocean circulation likely contribute to the temperature difference".

**4**. Throughout the manuscript: the authors refer to their idealized orbit (e.g., page 7, line 6) to name their '23.5° obliquity, null eccentricity' orbital configuration. The authors may refer to the "median orbit" used by De Vleeschouwer et al. (2014). Please also quickly justify this choice for the best-guess runs (minimal seasonality?).

**5**. Section 2.2.4 and Fig. 10: I do not get the age interval associated with each continental reconstruction. Why are the authors writing "410+-5Ma"? How is it defined? See also Fig. 9.

**6**. Page 7, line 22: "vegetation is spread globally". Is that true? After Section 2.2.2, it seems that areas located farther than 1500 km to the coast and regions higher than 1500 m in altitude are not covered.

**7**. Page 20, lines 22-25: Given the large margins of uncertainty associated with the model boundary conditions and atmospheric model (see major comments), I am not sure that it is reasonable to draw this kind of conclusions.

**8**. Page 22, lines 5-8: "Best-guess simulations […] show a general climatic cooling trend in accordance with reconstructions showing decreasing levels of atmospheric $CO_2$ concentrations during the Devonian". This is circular reasoning. The drop in $CO_2$ is a boundary condition of the model. On the contrary, the simulations do not capture the temperature trend reported based on proxy data throughout the Devonian (Fig. 10).

**9**. Table 1: Why are the authors not conducting their sensitivity test to vegetation cover on the 380 Ma configuration, like they did for the orbital parameters? Is there any reason for this?

**10**. Figure 1: It may be helpful to include stage names.

**11**. Figure 3: Continental outlines may be helpful. Also, the figure caption should clearly state how the difference is computed. I guess it is 415Ma–360Ma.

**12**. Figure 4: The figure is not easy to read. Would it be better using discrete color map levels and contours? Also, I suspect the increase in albedo in the NH shown in Fig. 4e (through sea-ice development obviously) to play an important role in the cooling shown in Fig. 4g. This may deserve a few words when discussing the patterns shown in Fig. 4g. Last point, the authors refer to the snow cover. Please show this on the maps.

**13**. Figure 9: Please state that the thick black line represents the coastline at the ocean resolution.

**14**. Figure 10: As it is, this figure is not very instructive. A way forward would be to investigate the SST changes simulated at the precise locations where the proxy data have been recovered. The simulated spread could be even larger when considering seasonal variations rather than (I guess, this is not stated) annual mean values.

---

## Referee Comment (RC2) · Anonymous Referee #2 · 17 May 2018

Reviewer 2 – May 2018

Brugger J., et al: On the sensitivity of the Devonian climate to continental configuration, vegetation cover and insolation

Recommendation: Major revision

Let me emphasize that this is an interesting study however the manuscript can be improved-in particular to make its importance clearer to the reader. In the present version, several issues are addressedÂă: (1) continental configuration, (2) vegetation cover and (3) the orbital forcing, but without to extract the major points for consideration. For instance, sections 3.3 and 3.4 present minor findings for the Devonian period, while

most significant contributions (sections 3.2 and 3.5) remain not enough explored. This problem being easily solvable, I recommend a major revision.

In addition to recommendations listed by the first reviewer I identified several areas requiring clarification.

Major Comments Ăă:

(1) The revised manuscript should provide a table showing exactly how vegetation types are parameterized. Surface albedo, roughness, and evapotranspiration coefficient values used for representing bare soil, shrub and tree have to be presented. It would be helpful to have a brief description of what evaporation/roughness is (in the model) because latent and sensible heat fluxes are both affected by these parameters. If relevant, the phenology should be discussed as well.

(2) The vegetation cover is never presented! Maps of vegetation used as boundary conditions for Middle and Late Devonian would be very helpful, especially for comparing with the figure 4. Moreover, as landplants are very sensitive to temperature-moisture regimes, it would be interesting to check if assumptions used to constrain the spreading of plants (shrub and tree) remain in good agreement with model's outputs.

(3) Personally, I'm skeptical about the interest of the section 3.4. The main reason is that the climatic effect remains very weak, so almost impossible to link with temperature estimates based on $\delta$18O, and potentially dependent on pCO2 levels. I suggest to remove this part, or significantly reduce its length.

(4) On lines 19-21 p 20. Authors argue that their results are in disagreement with Le Hir et al. 2011 findings. That is not entirely correct. Le Hir et al. 2011 suggested that the progressive change of the continental albedo has induced a warming (+4°C), but they have also noticed that this warming was not observed in their simulations due to the parallel reduction of the pCO2. Over the Devonian, the cooling was estimated to -1,9°C in response to the decreasing effectiveness of the greenhouse effect (carbon

dioxide level decreases from 6296 to 2125ppmv). To my knowledge, both studies only differ by their climate sensitivity ($\Delta T/\Delta pCO2$) to land cover change.

(5) A brief paragraph summarizing limitations of the model/study will be helpful for readers not familiar with models. For instance authors should take more cautions with their conclusions concerning the weak influence of the continental configuration - this result being mainly due to the absence of the climate-carbon feedback.

In addition to the above points, there are a number of minor errors that ought to be fixed:

- line 8 p9: For illustrating the impact of paleogeography, continental temperatures appear more relevant.

- the figure 4 is unreadable in its present state. How to compare Shrub-bare soil and Tree-shrub results ? please add panels showing Tree-Bare soil results. To make a more robust analysis, a plot of the snowline over continents should be included in surface albedo panels.

- line 10 p10: if you want to make that statement, a basic computation of the greenhouse effect may be helpful. (a simple formulation is available in Pierrehumbert 2005. (Climate dynamics of a hard snowball Earth, J. Geophys. Res., 110, D01111, doi:10.1029/2004JD005162.)

- line 14 p11: continental temperatures seem to be more relevant.

- on lines 1-4 p 14, authors conclude that "the discrepency ... we find that meridional ocean heat transport largely compensates for seasonal and regional differences in insolation caused by changes in orbital parameters." This result contrast with De Vleeschouwer et al. (2014) and constitutes an interesting finding of this study, so I suggest to include a specific discussion to convince the reader about the importance of the meridional heat transport (a figure will be very instructive).

- line 28 p18 ÂńÂă...increased precipitation . . . an increase in latent flux.ÂăÂż The

phrasing in this sentence is awkward. I am not sure that it is reasonable to mention this process to explain a warming at the surface.

---

## Author Comment (AC1) · 17 May 2018

**Response to Reviewer 1's comments**

First of all we would like to thank the reviewer for the very thorough report, the constructive criticism and for the numerous helpful suggestions which will certainly help to improve the presentation of our paper. We do not agree, however, with the overall recommendation to reject our paper since the comments can be thoroughly addressed during a revision of the manuscript as detailed below.

Recommendation: Reject.

Anonymous

Paper summary:

Brugger et al. use the ocean-atmosphere model of intermediate complexity Climber-3$\alpha$ to quantify the sensitivity of the Devonian climate system to changes in the configuration of the continents, in vegetation cover and in orbital parameters. They also analyze patterns of quasi-periodic variations in global mean surface temperature arising from the waxing and waning of sea ice over the North Pole. In a 3rd step, they combine changes in the above-mentioned boundary conditions, plus $pCO_2$ and solar constant, to conduct best-guess simulations, the purpose of which is to best capture the climatic conditions during the Early, Middle and Late Devonian (415, 380 and 360 Ma respectively). They focus their analysis on changes in surface air temperatures (SAT).

The authors demonstrate a weak impact of the paleogeographical evolution on globally-averaged SAT, although strong seasonal changes arise at the regional scale in response to the changing landsea mask. Changes in vegetation cover, from bare land to coastal shrub and then trees, impact simulated mean annual SAT by no more than 0.2 °C. Over continents, the warming induced by the decrease in albedo is overwhelmed by the increase in evaporation and associated cooling effect. The main signal emerging from the sensitivity tests to the orbital configuration is the increase in global temperature with increasing obliquity and increasing eccentricity, whereas precession has a very minor impact. Regarding the analysis of the periodic flips between two climatic states over the North Pole, the authors suggest a mechanism linking sea-ice extent, ocean salinity and convective processes. The resulting climatic mechanism is associated with changes in global SAT on the order of 0.2 °C. Best-guess model runs feature a cooling trend throughout the Devonian, with mean annual globally-averaged SAT decreasing from 22.2 °C in the Early Devonian to 21.0 °C in the Middle Devonian and 19.3 °C in the Late Devonian. The authors explain that this trend is mainly due to the imposed drop in $pCO_2$ throughout the period (from 2000 ppmv to 1000 ppmv), while the impact of changes in vegetation cover can be neglected. These results contrast with a previous study that suggested that the climatic impact of decreasing $pCO_2$ throughout the Devonian may

have been compensated by the concomitant decrease in albedo induced by the spread of land plants (Le Hir *et al.*, 2011). Simulated tropical sea-surface temperatures are systematically lower than suggested by proxy data. Brugger et al. therefore suggest that the ocean $\delta 18O$ composition may have been lower in the Devonian, thus biasing temperature reconstructions.

Major comments:

In this contribution, the authors conduct numerous sensitivity tests. They cover several topics previously investigated by other research groups using other climate models, namely the sensitivity to the vegetation cover and to the orbital configuration (Le Hir *et al.*, 2011; De Vleeschouwer *et al.*, 2014). As stated by the authors in their introduction, this is important to determine the response of the Devonian climate system to different kinds of perturbations, in order to better understand the mechanisms potentially lying behind the coupled changes in the ocean-atmosphere system and in the biosphere at that time. Such systematic experiments should also permit to determine to what extent the results obtained in previous studies are model-dependent. The manuscript is concise and well-written and the experimental setup, in spite of the numerous experiments and sensitivity tests, is well described so that the study is easy to follow. This is appreciated.

Thank you, we have spent considerable time and effort to make it easier for the reader to navigate through the wealth of material presented in our manuscript.

However, the paper does not convey any clear, robust message. The authors cover a wide spectrum of scientific questions and provide numerous experiments, but this generally goes with a lack of robust analysis of the model output. For this reason and other weaknesses exposed in detail hereafter, I do not support publication of the manuscript in Climate of the Past. Below, I explain my motivations in rejecting this manuscript and I try to propose constructive comments.

We disagree that our paper does not convey any clear message, see the points clearly listed in Section 4 of our manuscript. We do appreciate, however, the detailed suggestions for more robust analysis and additional sensitivity studies which will be included in the revised version of our manuscript, see our more detailed responses below.

1. On the choice of the numerical model.

I do not like recommending using up-to-date, highresolution models. Climate models of previous generations are useful because they generally provide results that are very close at first order to the ones obtained using more robust models while permitting numerous sensitivity tests due to their lower computational cost. This is obviously what motivated Brugger et al. to use Climber-$3\alpha$. However, I am wondering if this ocean

GCM coupled with a very coarse-resolution statistical-dynamical atmosphere model is really adapted to the kind of experiments the authors are conducting. Indeed, most of the signal analyzed by the authors is provided by the atmospheric component, be it in response to changing vegetation cover or orbital configuration. In that context, I would suggest adopting a relatively robust atmospheric model. I do not necessarily want to say that the authors should repeat their experiments with another model, but please, at least, clearly state the name of the model rather than hiding this behind a reference to Montoya et al. (2005) and include a discussion of the limitations. I also suggest clearly quantifying the bias associated with the interpolation onto the $22.5° \times 7.5°$ grid, see below.

While it is true that part of the effects are clearly driven by the atmosphere, we do point out the importance of the ocean, in particular with respect to changes in orbital configuration. In fact, this is one of the key results in our paper. We also would like to point out that in earlier studies of climate problems over a wide range of different time intervals results from our model do agree very well with studies employing a more sophisticated atmosphere model, e.g. for simulations for the 21st century (Feulner & Rahmstorf 2010 later confirmed by Anet *et al.* 2013; Meehl *et al.* 2013; Ineson *et al.* 2015; Maycock *et al.* 2015; Chiodo *et al.* 2016), for the last millennium (Feulner 2011 confirmed by Schurer *et al.* 2014) or for Neoproterozoic glaciations (Feulner & Kienert 2014 in very good agreement with Liu *et al.* 2013).

Thus, in our opinion, the reviewer is certainly correct in the assumption that our model is "very close at first order to the ones obtained using more robust models". We agree, however, that we need to discuss potential model limitations more thoroughly and will do so in the revised version of the manuscript, see also our comment on the coarse resolution further below. Finally we would like to point out clearly that the rationale behind the omission of the model name was certainly not to "hide" anything but to avoid the frequent confusion between the different versions of the CLIMBER model. We will include the model name in the revised text.

2. On the weakness of the signal identified in the Arctic.

The manuscript is generally well organized: the authors first investigate the impact of different forcing factors on simulated Devonian climate and subsequently combine these parameters to conduct their best-guess model runs. Section 3.4 Flips between two climatic states in the Arctic does not fit within this framework. It impedes the reading flow and, more critically, the amplitude of the SAT signal identified by the authors ca. $0.2°C$ is so weak that it would have no significant implications on the deep-time sedimentary record. I suggest deleting this part of the manuscript, which would at the same time allow the authors to expand on the other topics and thus to make their manuscript more robust in many aspects, see other major comments.

While Section 3.4 may not be ideal in terms of text flow, we object to the recommendation to delete it from the manuscript for several reasons. First, the regional effect around the North pole of this (to our knowledge) novel climate variability pattern is very pronounced despite the fact that the global signal is small. Although the global temperature signal is small ($\sim$0.2°C), surface air temperature fluctuations over the Arctic region are on the order of 1.5°C and reach $\sim$5.5°C at the North pole. Regional temperature changes are therefore not negligible. Second, the waxing and waning of winter sea-ice cover with a periodicity of several centuries has the potential to impact the Arctic marine ecosystems considerably. In a recent work, Harada (2016) documents tremendous changes in the biogeochemical cycling due to the anthropogenic climate change in the western Arctic Ocean including all trophic levels. Third, it is an interesting mechanism in itself which might be operating during other time periods as well and thus should be described in the scientific literature. In summary we consider this Section an integral and important part of our manuscript. We will, however, consider re-ordering Sections 3.4 and 3.5 to improve the text flow.

3. On the lack of sensitivity tests to the $pCO_2$.

The rationale adopted by Brugger et al. more or less consists in decomposing the signal obtained in their best-guess model runs through individual sensitivity tests to the different parameters in play (continental configuration, vegetation cover). In this context, it seems that a sensitivity test to the $pCO_2$ would be helpful. In particular, the authors state on page 18, lines 17-19 that The global cooling over the Devonian seen in the set of best-guess simulations [. . . ] is dominantly driven by the strong decrease in $CO_2$ concentrations. This statement is not supported by the results provided in the current version of the manuscript. One could further imagine a series of best-guess experiments with $pCO_2$ held constant in order to clearly show that this constitutes the main driving factor of simulated climate cooling. The same could be done for the solar constant, the impact of which is not independently quantified. Also, the impact of changes in the orbital parameters is tested in Section 3.3 but it is not subsequently discussed. For instance, one could imagine showing the SST spread associated with different orbital configurations in Fig. 10.

We have inferred the sensitivity to $CO_2$ indirectly from the rather limited response to other factors, but the reviewer is certainly correct in pointing out the lack of dedicated sensitivity experiments with respect to changes in $pCO_2$ and the solar constant alone. We will be happy to provide an analysis of additional sensitivity experiments along these lines in the revised version of our paper.

4. On the need for a more robust analysis of the impact of the paleogeographical changes.

In Section 3.1, the authors explain that changes in the land-sea mask induce strong seasonal SAT changes at the regional scale. What about the changes in land surface

area and associated surface albedo? Please provide a table including key numbers such as land surface area for each timeslice. This is particularly important because results of the interpolation of the paleomaps shown in Fig. 2 onto the coarse grid of the atmospheric model are uncertain. Please check that the surface area on this grid is in firstorder agreement with the original reconstructions provided by C. Scotese and plotted in Fig. 2.

We think that this concern may be partly due to a misunderstanding: The (admittedly very coarse) atmospheric model operates in a way that fractions of land (or vegetation for that matter) within the comparatively large grid cells are taken into account when computing the albedo, the evaporation or the surface energy balance, so the bias due to the spatial resolution of our atmospheric grid should not be large. However, we will quantify and discuss this more thoroughly in the revised version of the paper.

5. On the need for a more robust analysis of the vegetation cover impact.

The authors show that, in their simulations, the spread of Devonian land plants induces no critical changes in the mean annual, globally-averaged SAT. These results are interesting because they contrast with previous work (Le Hir *et al.*, 2011), which suggested that the decrease in albedo may have been sufficient to counterbalance the drop in $CO_2$. I think that this may constitute the most interesting part of the manuscript. However, several points need to be discussed/tested before drawing any robust conclusions:

We appreciate the reviewer's suggestion to turn this part of our sensitivity study into a more central part of the manuscript. As discussed in depth below, the limited knowledge of early land plant evolution might make a very detailed investigation difficult, but we are certainly willing to considerably strengthen the analysis presented in this part of our paper.

Unknown vegetation cover. Maps of imposed vegetation cover are not provided and the surface area occupied by these plants in the different experiments (i.e., coastal shrub and trees) is not specified. In other words, the boundary conditions are unknown. The authors describe how they built their maps in Section 2.2.2, but this remains obscure. For instance, they explain that they assume 80 % grass-like vegetation cover in areas lower than 500 m in altitude and closer than 500 km to the coast. Well, what does it give on the 22.5°x7.5° grid? Whats the magnitude of the bias (in terms of surface area) imposed by the interpolation onto this grid?

We fully agree with the reviewer that the manuscript would profit from a graphical presentation of the vegetation cover used in our sensitivity experiments to give readers a better idea of the input vegetation distributions. Appropriate maps will be included in the revised version. As already mentioned above, the bias due to the coarse model resolution should not be large, because the fraction of land inside a grid cell is taken into

account when calculating vegetation cover and its impact on climate. We will discuss this and the potential impact on our results in the revised version of the paper.

Lack of justification for the definition of the plant spatial cover. Also, the authors do not justify their choices regarding the imposed land plant cover: why are trees absent in continental interiors? For instance, Le Hir *et al.* (2011) used a vegetation model and trees are simulated over the South Pole. Similarly, why is vegetation absent beyond 1500 m in altitude? This discussion is of first-order importance, since the choices made here drive an essential part of the model response to changing vegetation cover throughout the period.

In general, it was our intention to model the changes in vegetation based on very simple, basic assumptions. Although there is much literature on the evolution of land plants before and during the Devonian, there is, in our opinion, still insufficient knowledge to justify the use of a more complex vegetation model.

The assumptions behind our scenarios for the expanding spatial distribution of vascular land plants during the Devonian are based on empirical evidence for land plant evolution during that time period as discussed in Algeo *et al.* (1995). In contrast to later time periods, however, we do not have enough information to reconstruct a detailed world map of vegetation cover, but have to use a simpler approach based on quantities like distance from the coast or ground elevation which can be considered as proxies for physiological limits of vegetation growth.

For example, before the Late Devonian, early vascular plants are thought to have been restricted to regions close to water due to the absence of deep roots (Algeo *et al.*, 1995), effectively leading to a limit in terms of distance from the coast (page 6, lines 23-33 in our paper). Furthermore, vascular plants throughout the Devonian may have been confined to lowlands for similar reasons (Boyce & Lee, 2017), translating into an additional limit in terms of elevation. Finally, trees only appeared during the Late Devonian (Algeo *et al.*, 1995), and the evolution of deeper roots allowed for a more extended distribution of vascular plants.

We agree that these assumptions are not explained in sufficient detail and will fix this issue in the revised version. Furthermore, although the distances from the coast, the altitude to which we limit plants and the fractions covered by shrub-like vegetation and trees have been chosen to approximately represent our current knowledge of land plant evolution as outlined above, there is certainly some freedom in these parameter choices. Therefore we agree with the reviewer that additional sensitivity experiments are required to assess the impact of these parameters on our results.

Lack of justification for the definition of the plant expansion scenario. Closely related to my previous point, I see no robust justification for the scenario imposed by the authors in their best-guess model runs. Le Hir *et al.* (2011) conducted an interesting review of the

literature to propose a scenario of Devonian land plant evolution in their models. They identified several steps in the colonization of the land surface by Devonian plants and selected appropriate plant functional types accordingly for each time slice. The same kind of approach is required to ensure robust modeling results. I am particularly surprised that the authors impose a bare land in their Early Devonian best-guess model configuration: it is recognized that non-vascular plants appeared as soon as the Mid Ordovician (Rubinstein et al., New Phytologist 2010) and that vascular plants were already present by the Late Ordovician (Steemans et al., Science 2009), with possibly strong implications on the Earth System at that time (Lenton et al., NatGeo 2012). Modeling studies further suggest that non-vascular plant cover may have been relatively widespread during the Ordovician (Porada et al., NatCom 2016; Lenton et al., PNAS 2016). I encourage the authors to explore the literature to build their experimental design more robustly.

We appreciate the recommendation of consulting additional literature in order to improve the scenarios for land-plant evolution throughout the Devonian. There are two important points here. First, based on the literature, we agree that we should take into account that non-vascular plants in the Early Devonian were likely already globally distributed and will modify our baseline scenario accordingly. Second, we will adjust the properties of plant types (in particular albedo, evapotranspiration and roughness length) and their change through the Devonian based on Le Hir *et al.* (2011) and the literature cited therein by representing Devonian plants by the most suitable modern analog (Matthews, 1984; Hack *et al.*, 1993).

Poorly documented plant properties. The properties of the vegetation cover are poorly documented. What are the values of evapotranspiration and roughness associated with each plant functional type in use? Changes in evapotranspiration are a primary driver of the simulated SAT changes. Values have to be discussed.

This is a very important point, in particular since we intend to vary those parameters (albedo, evapotranspiration, roughness length) in our revised version to better represent the evolution of land plants during the Devonian (see our response to the previous comment). All values will be documented and discussed in the revised version.

Lack of justification for the plant properties. Also, do the plant properties correctly represent Devonian plants? The authors should discuss this and use their numerical model to investigate the range of possible values, in order to determine if their results are robust. This is a crucial point.

In the revised version, we will discuss plausible values for plant properties during the Devonian. As the information about Devonian plant properties is scarce, we will perform and discuss sensitivity experiments exploring the impact of evapotranspiration on our results in addition to the tests with respect to albedo values described in the current version of the paper (page 11, lines 11-20).

Minor comments:

1. Title of the manuscript: I suggest replacing insolation with orbital configuration to make it clear that this does not refer to the changes in solar constant, which are also accounted for in the bestguess model runs.

We will change the title in the revised version.

2. Page 4, lines 20-21: Changes in solar constant are not systematically investigated.

As outlined above, additional sensitivity experiments will be conducted in order to investigate changes in the solar constant with all other variables remaining fixed.

3. Please avoid throwaway sentences, e.g. on page 9, lines 12-13: Furthermore, shifts in the distribution of land and ocean areas as well as changes in ocean circulation likely contribute to the temperature difference.

This will be more thoroughly analysed in the revised version.

4. Throughout the manuscript: the authors refer to their idealized orbit (e.g., page 7, line 6) to name their 23.5° obliquity, null eccentricity orbital configuration. The authors may refer to the median orbit used by De Vleeschouwer et al. (2014). Please also quickly justify this choice for the best-guess runs (minimal seasonality?).

We will add a justification during revision and will consider renaming our standard orbital configuration.

5. Section 2.2.4 and Fig. 10: I do not get the age interval associated with each continental reconstruction. Why are the authors writing 410+-5Ma? How is it defined? See also Fig. 9.

The uncertainty in time was meant to reflect uncertainties in the reconstructions of continental configurations and in the range of $CO_2$ concentrations, but we fully acknowledge that this is somewhat confusing. We will change this during revision.

6. Page 7, line 22: vegetation is spread globally. Is that true? After Section 2.2.2, it seems that areas located farther than 1500 km to the coast and regions higher than 1500 m in altitude are not covered.

True, we meant to say that it is more widespread; this will be changed during revision.

7. Page 20, lines 22-25: Given the large margins of uncertainty associated with the

model boundary conditions and atmospheric model (see major comments), I am not sure that it is reasonable to draw this kind of conclusions.

As explained above, we do not think that we are limited by our atmospheric model, but the point regarding the uncertainties associated with boundary conditions is well taken. We will rephrase this sentence to reflect the uncertainties. Furthermore, we will include a more thorough discussion of potential model limitations in the revised version of the paper.

8. Page 22, lines 5-8: Best-guess simulations [...] show a general climatic cooling trend in accordance with reconstructions showing decreasing levels of atmospheric $CO_2$ concentrations during the Devonian. This is circular reasoning. The drop in $CO_2$ is a boundary condition of the model. On the contrary, the simulations do not capture the temperature trend reported based on proxy data throughout the Devonian (Fig. 10).

Many thanks for spotting this, but it is a matter of misleading wording rather than circular reasoning. What we meant to say is that the combined effect of all boundary conditions leads (in our model at least) to a general cooling which appears to be dominated by the falling $CO_2$ concentrations as derived from reconstructions. (Note that we will provide dedicated sensitivity experiments investigating the effects of $CO_2$ alone to make this statement more robust.) The reviewer is correct in pointing out that, similar to other studies the cooling seen in our model does not match the reconstructed temperature trend. A discussion of potential uncertainties and model limitations will be added to the revised text.

9. Table 1: Why are the authors not conducting their sensitivity test to vegetation cover on the 380 Ma configuration, like they did for the orbital parameters? Is there any reason for this?

The main argument for choosing the 360 Ma continental configuration for the sensitivity simulations with respect to changing vegetation cover was that trees appeared only very late in the Devonian. While we expect the choice of continental configuration to have only a minor influence on the results, we do acknowledge that it makes more sense to do all sensitivity experiments on the same continental configuration. We will therefore conduct the improved vegetation sensitivity study with the 380 Ma continental configuration for the revised version of the paper.

10. Figure 1: It may be helpful to include stage names.

Good point, stage names will be added to Figure 1 in the revised version.

11. Figure 3: Continental outlines may be helpful. Also, the figure caption should clearly state how the difference is computed. I guess it is 415Ma-360Ma.

The problem with continental outlines is that they change between the two time periods for which we subtract the temperatures in this Figure; we will try out a version with different contours for both time intervals. Furthermore, the caption will clearly state how the difference is computed.

12. Figure 4: The figure is not easy to read. Would it be better using discrete color map levels and contours? Also, I suspect the increase in albedo in the NH shown in Fig. 4e (through sea-ice development obviously) to play an important role in the cooling shown in Fig. 4g. This may deserve a few words when discussing the patterns shown in Fig. 4g. Last point, the authors refer to the snow cover. Please show this on the maps.

We will try to improve this figure as suggested by the reviewer, in particular the points about sea-ice and snow cover are important, so we will try to include this information in the figure.

13. Figure 9: Please state that the thick black line represents the coastline at the ocean resolution.

This information will be added during revision.

14. Figure 10: As it is, this figure is not very instructive. A way forward would be to investigate the SST changes simulated at the precise locations where the proxy data have been recovered. The simulated spread could be even larger when considering seasonal variations rather than (I guess, this is not stated) annual mean values.

It is a good idea to look into the seasonal spread as well, we will investigate this issue during revision. We are not certain about the reviewer's suggestion to use the model data at the precise location of the proxy records. In fact, we had thought about this option but decided against it because the proxy record at one specific location could be influenced by local effects not necessarily captured in our relatively coarse resolution model. Since the proxy time series consists of two records at 10–15°S and three records at 25–30°S, we considered the spread of model temperatures between those latitude bands a more robust measure for the comparison. However, we will check how this choice impacts the temperature ranges shown in the figure and will discuss this point in more detail in the revised version.

**References**

Algeo, T. J., Berner, R. A., Barry, M. J. & Scheckler, S. E. 1995: *Late Devonian Oceanic Anoxic Events and Biotic Crises: "Rooted" in the Evolution of Vascular Land Plants?*, GSA Today, 5, 45,64

Anet, J. G., Rozanov, E. V., Muthers, S., Peter, T., Brönnimann, S., Arfeuille, F., Beer, J., Shapiro, A. I., *et al.* 2013: *Impact of a potential 21st century "grand solar minimum" on surface temperatures and stratospheric ozone*, Geophys. Res. Lett., 40, 4420

Boyce, C. K. & Lee, J.-E. 2017: *Plant Evolution and Climate Over Geological Timescales*, Annual Review of Earth and Planetary Sciences, 45, 61

Chiodo, G., García-Herrera, R., Calvo, N., Vaquero, J. M., Ael, J. A., Barriopedro, D. & Matthes, K. 2016: *The impact of a future solar minimum on climate change projections in the Northern Hemisphere*, Environ. Res. Lett., 11, 034015

De Vleeschouwer, D., Crucifix, M., Bounceur, N. & Claeys, P. 2014: *The impact of astronomical forcing on the Late Devonian greenhouse climate*, Global Planet Change, 120, 65

Feulner, G. 2011: *Are the most recent estimates for Maunder Minimum solar irradiance in agreement with temperature reconstructions?*, Geophys. Res. Lett., 38, L16706

Feulner, G. & Kienert, H. 2014: *Climate simulations of Neoproterozoic snowball Earth events: Similar critical carbon dioxide levels for the Sturtian and Marinoan glaciations*, Earth Planet. Sc. Lett., 404, 200

Feulner, G. & Rahmstorf, S. 2010: *On the effect of a new grand minimum of solar activity on the future climate on Earth*, Geophys. Res. Lett., 37, L05707

Hack, J. J., Boville, B. A., Briegleb, B. P., Kiehl, J. T. & Williamson, D. L. 1993: *Description of the NCAR Community Climate Model (CCM2), NCAR Technical Note*, National Center for Atmospheric Research, Boulder, Colorado

Harada, N. 2016: *Review: Potential catastrophic reduction of sea ice in the western Arctic Ocean: Its impact on biogeochemical cycles and marine ecosystems*, Global and Planetary Change, 136, 1

Ineson, S., Maycock, A. C., Gray, L. J., Scaife, A. A., Dunstone, N. J., Harder, J. W., Knight, J. R., Lockwood, M., *et al.* 2015: *Regional climate impacts of a possible future grand solar minimum*, Nat. Commun., 6, 7535

Le Hir, G., Donnadieu, Y., Godderis, Y., Meyer-Berthaud, B., Ramstein, G. & Blakey, R. C. 2011: *The climate change caused by the land-plant invasion in the Devonian*, Earth Planet Sc Lett, 310, 203

Liu, Y., Peltier, W. R., Yang, J. & Vettoretti, G. 2013: *The initiation of Neoproterozoic "snowball" climates in CCSM3: the influence of paleocontinental configuration*, Clim. Past, 9, 2555

Matthews, E. 1984: *Prescription of Land-Surface Boundary Conditions in GISS GCM II: A Simple Method Based on High-Resolution Vegetation Data Bases, NASA Technical Memorandum 86096*, National Aeronautics and Space Administration, Goddard Space Flight Center Institute for Space Studies, New York

Maycock, A. C., Ineson, S., Gray, L. J., Scaife, A. A., Anstey, J. A., Lockwood, M., Butchart, N., Hardiman, S. C., *et al.* 2015: *Possible impacts of a future grand solar minimum on climate: Stratospheric and global circulation changes*, J. Geophys. Res., 120, 9043

Meehl, G. A., Arblaster, J. M. & Marsh, D. R. 2013: *Could a future "Grand Solar Minimum" like the Maunder Minimum stop global warming?*, Geophys. Res. Lett., 40, 1789

Schurer, A. P., Tett, S. F. B. & Hegerl, G. C. 2014: *Small influence of solar variability on climate over the past millennium*, Nature Geoscience, 7, 104

---

## Short Comment (SC1) · 18 May 2018

Brugger et al. present several Devonian paleoclimate simulations with a coupled Earth system model of intermediate complexity. The authors describe and compare different steady-state experiments, characterized by different continental configurations, solar constant, CO2 concentration and orbital configuration. In this short comment, I focus on their sensitivity analysis of Devonian climate (at 380 Ma) to orbital forcing.

Brugger et al. document a very low sensitivity of the Devonian climate to orbital forcing, reporting only 0.6 degrees C difference in mean annual global temperature between their coldest and warmest simulation. In a similar sensitivity analysis, my coauthors and I found a 7 degrees difference in mean annual global temperature between the coldest and warmest orbit (De Vleeschouwer et al., 2014, Global and Planetary Change). The authors attribute this order of magnitude discrepancy "to the slab ocean model used by De Vleeschouwer et al. (2014)" in contrast to the dynamical ocean model that is used in their study, in which meridional ocean heat transport can vary on orbital time-scales. I agree with the authors that this difference in model setup plays a role and could explain part of the discrepancy. However, the authors do not mention potential differences in (continental) snow-albedo positive feedback mechanisms on the Gondwanan continent, which in my opinion is far more important. The extent of the seasonal snow cover on the Gondwanan continent turned out to be a major and powerful positive feedback mechanism in the De Vleeschouwer et al. (2014) simulations. However, this important aspect of the climate system is not discussed in the manuscript by Brugger et al. Hence, I strongly suggest that the authors look into the response of continental snow and ice to orbital forcing, and discuss their role in the Devonian climate system.

I am also surprised by the perfect symmetry in Figure 5. This symmetry implies that precession does not influence mean annual global temperature at all. This is –to say the least- unexpected given the asymmetry in continental configuration and the large difference in heat capacity between oceans and continents. The authors recognize the importance of an asymmetrical continental configuration on page 13, writing "the strong southward shift of the continents amplifies the stronger seasonal insolation at the South Pole for higher obliquity values, resulting in higher global annual mean surface temperatures". If this is true for obliquity, one should ask why the model does not simulate higher global annual mean surface temperatures when the precessional configuration is such that the Earth is close to the Sun (at perihelion) during austral summer. In order to be able to better assess this remarkable result, I would like to see a map showing the surface temperature difference between a precession maximum and minimum (difference between [e = 0.069; $\omega$ = 90] and [e = 0.069, $\omega$ = 270]) for JJA, DJF and the annual mean.

Minor comment:

Page 4, line 6. The total amount of insolation received by the Earth over a year varies only to a very small extent with eccentricity forcing. Changes in the spatial and temporal distribution of insolation is by far the more important aspect of orbital forcing. I would suggest to mention those in order of importance.

---

## Author Comment (AC2) · 10 Jul 2018

**Response to Reviewer 2's comments**

First we would like to thank the reviewer for acknowledging the relevance of our results and for his suggestions. This will certainly help to improve our manuscript considerably.

Recommendation: Major revision.

Anonymous

Let me emphasize that this is an interesting study however the manuscript can be improved-in particular to make its importance clearer to the reader. In the present version, several issues are addressed: (1) continental configuration, (2) vegetation cover and (3) the orbital forcing, but without to extract the major points for consideration. For instance, sections 3.3 and 3.4 present minor findings for the Devonian period, while most significant contributions (sections 3.2 and 3.5) remain not enough explored. This problem being easily solvable, I recommend a major revision.

We agree with the reviewer that our studies' relevance needs to be articulated more clearly. In our opinion this definitely includes Section 3.3 and 3.4: Several papers convincingly argue that the influence of the orbital configuration (Section 3.3) is an important aspect for the Devonian period (De Vleeschouwer *et al.*, 2014, 2017). Furthermore, we find the climate variability pattern described in Section 3.4 of importance, as this strong regional effect might be able to impact marine biogeochemistry and therefore marine ecosystems at high Northern latitudes (Harada, 2016). For the Devonian as a period of several oceanic mass extinctions, we therefore see this as a crucial aspect to discuss.

The influence of changes in vegetation cover (Section 3.2) will be explored in greater depth in the revised version (see also our response to the first reviewer's comments). In particular, we will address the following aspects:

- A graphical presentation of the vegetation distributions used in the model experiments will be added.

- We will better explain the assumptions made for the vegetation cover of the different timeslices and add sensitivity experiments to better assess the uncertainty of the parameter choices made.

- The global distribution of non-vascular plants in the Early Devonian will be taken into account.

- Values of the most important vegetation parameters (albedo, evapotranspiration, roughness length) and their variation will be listed.

Finally, we will revise Section 3.5 according to the suggestions of both reviewers.

In addition to recommendations listed by the first reviewer I identified several areas requiring clarification.

Major comments:

(1) The revised manuscript should provide a table showing exactly how vegetation types are parameterized. Surface albedo, roughness, and evapotranspiration coefficient values used for representing bare soil, shrub and tree have to be presented. It would be helpful to have a brief description of what evaporation/roughness is (in the model) because latent and sensible heat fluxes are both affected by these parameters. If relevant, the phenology should be discussed as well.

We appreciate this suggestion and will provide the information in the revised version of the paper.

(2) The vegetation cover is never presented! Maps of vegetation used as boundary conditions for Middle and Late Devonian would be very helpful, especially for comparing with the figure 4. Moreover, as landplants are very sensitive to temperature-moisture regimes, it would be interesting to check if assumptions used to constrain the spreading of plants (shrub and tree) remain in good agreement with models outputs.

As mentioned above, we will include maps of the vegetation distribution in the revised version. Additionally, we will have a look at the temperature-moisture regimes and how this fits our vegetation distributions.

(3) Personally, I'm skeptical about the interest of the section 3.4. The main reason is that the climatic effect remains very weak, so almost impossible to link with temperature estimates based on $\delta^{18}$O, and potentially dependent on pCO$_2$ levels. I suggest to remove this part, or significantly reduce its length.

As already outlined shortly above and motivated in detail in the reply to Reviewer 1, we are convinced of the importance of Section 3.4. The strong regional effect is dynamically interesting in its own right, and the relevance of the Arctic's biogeochemistry for marine life (Harada, 2016) makes the described mechanism important in the context of the oceanic mass extinctions during the Devonian.

(4) On lines 19-21 p 20. Authors argue that their results are in disagreement with Le Hir et al. 2011 findings. That is not entirely correct. Le Hir et al. 2011 suggested that the progressive change of the continental albedo has induced a warming (+4°C), but they have also noticed that this warming was not observed in their simulations due to the parallel reduction of the pCO$_2$. Over the Devonian, the cooling was estimated to -1.9°C in response to the decreasing effectiveness of the greenhouse effect (carbon dioxide level decreases from 6296 to 2125 ppmv). To my knowledge, both studies only differ by

their climate sensitivity ($\Delta T/\Delta pCO_2$) to land cover change.

We agree that the phrasing in this paragraph is misleading. The reviewer is correct in pointing out that Le Hir *et al.* (2011) report a temperature decrease of 1.9°C from their Early Devonian no-land-plants simulation to their Late Devonian simulation with Late Devonian land plant cover (called LP3 in Le Hir *et al.* 2011). This is a scenario comparable to our Early and Late Devonian best-guess simulations; using the same albedo values as in Le Hir *et al.* (2011) we find a 2°C temperature decrease, in very good agreement with their result.

However, while Le Hir *et al.* (2011) emphasise that the cooling due to falling $CO_2$ levels is mostly compensated by land-plant evolution (see, for example, their abstract where they report "unchanged temperatures"), we wanted to stress the fact that both studies find a cooling trend which is in contrast to the global warming seen in proxy data for the Late Devonian (Joachimski *et al.*, 2009; van Geldern *et al.*, 2006). In the revised version, we will clarify this issue.

(5) A brief paragraph summarizing limitations of the model/study will be helpful for readers not familiar with models. For instance authors should take more cautions with their conclusions concerning the weak influence of the continental configuration - this result being mainly due to the absence of the climate-carbon feedback.

Although we tried to mention the limitations of our model and our study in the relevant paragraphs, we agree that adding a paragraph which summarises these limitations is a good idea.

In addition to the above points, there are a number of minor errors that ought to be fixed:

line 8 p9: For illustrating the impact of paleogeography, continental temperatures appear more relevant.

We will add a discussion on the variations of continental temperatures with changing continental configuration.

the figure 4 is unreadable in its present state. How to compare Shrub-bare soil and Tree-shrub results ? please add panels showing Tree-Bare soil results. To make a more robust analysis, a plot of the snowline over continents should be included in surface albedo panels.

The intention of presenting the differences of the coastal-shrub minus the bare-soil experiments and the tree minus coastal-shrub experiments was to trace the evolution of land plants chronologically from the Early to the Middle Devonian and from the Middle to the Late Devonian in order to understand the associated changes in surface

air temperature. If space permits, we will show and discuss the difference between the tree and the bare-soil cases as well. Finally, adding the snowline in the surface albedo panels is a good idea in principle, but complicated in a difference plot due to the fact that the snowline is different for the various vegetation distributions. Instead, we propose to add snow and sea-ice cover to the surface-air temperature maps shown in Figure 9.

line 10 p10: if you want to make that statement, a basic computation of the greenhouse effect may be helpful. (a simple formulation is available in Pierrehumbert 2005. (Climate dynamics of a hard snowball Earth, J. Geophys. Res., 110, D01111, doi:10.1029/2004JD005162.)

Many thanks for this suggestion. We will explore this possibility, but also the option of supporting our statement more directly using model diagnostics.

line 14 p11: continental temperatures seem to be more relevant.

We will add continental temperatures, but also give surface air temperatures to ensure the comparability of our sensitivity simulations with each other.

on lines 1-4 p 14, authors conclude that the discrepency ... we find that meridional ocean heat transport largely compensates for seasonal and regional differences in insolation caused by changes in orbital parameters. This result contrast with De Vleeschouwer et al. (2014) and constitutes an interesting finding of this study, so I suggest to include a specific discussion to convince the reader about the importance of the meridional heat transport (a figure will be very instructive).

Triggered by the short comment by David De Vleeschouwer, we have now investigated this aspect and the comparison with their results more thoroughly. We are grateful that they supplied us with some of their model output data sets. Therefore, we can now add a paragraph discussing the causes of this discrepancy in more detail.

line 28 p18 ...increased precipitation ... an increase in latent flux. The phrasing in this sentence is awkward. I am not sure that it is reasonable to mention this process to explain a warming at the surface.

We agree. In the revised version we will focus in the description of the regional patterns on the shift of the intertropical convergence zone and its effect on temperature. We will discuss the influence of the interaction of the continents' distribution, the orography and the seasonality on this shift.

**References**

De Vleeschouwer, D., Crucifix, M., Bounceur, N. & Claeys, P. 2014: *The impact of astronomical forcing on the Late Devonian greenhouse climate*, Global Planet Change, 120, 65

De Vleeschouwer, D., Da Silva, A.-C., Sinnesael, M., Chen, D., Day, J. E., Whalen, M. T., Guo, Z. & Claeys, P. 2017: *Timing and pacing of the Late Devonian mass extinction event regulated by eccentricity and obliquity*, Nat Commun, 8:2268

Harada, N. 2016: *Review: Potential catastrophic reduction of sea ice in the western Arctic Ocean: Its impact on biogeochemical cycles and marine ecosystems*, Global and Planetary Change, 136, 1

Joachimski, M., Breisig, S., Buggisch, W., Talent, J., Mawson, R., Gereke, M., Morrow, J., Day, J., *et al.* 2009: *Devonian climate and reef evolution: Insights from oxygen isotopes in apatite*, Earth Planet Sc Lett, 284, 599

Le Hir, G., Donnadieu, Y., Godderis, Y., Meyer-Berthaud, B., Ramstein, G. & Blakey, R. C. 2011: *The climate change caused by the land-plant invasion in the Devonian*, Earth Planet Sc Lett, 310, 203

van Geldern, R., Joachimski, M., Day, J., Jansen, U., Alvarez, F., Yolkin, E. & Ma, X.-P. 2006: *Carbon, oxygen and strontium isotope records of Devonian brachiopod shell calcite*, Palaeogeogr Palaeocl, 240, 47

---

## Author Comment (AC3) · 10 Jul 2018

**Response to David De Vleeschouwer's comment**

We thank David De Vleeschouwer for his comments considering the comparison of our sensitivity analysis of the Devonian climate to orbital forcing with results of their study (De Vleeschouwer *et al.*, 2014). In this response, we will provide the maps he had asked for in his comment and several other figures which help to better understand the discrepancy between their and our results. We also thank him for making their data for the heat flux used for calibration as well as the sea-ice data for the median orbit and the minimum and maximum obliquity simulations available to us.

Hereafter, we first have a look at the response of continental snow and sea ice to orbital forcing, as suggested by David De Vleeschouwer. We will then give a short comparison of the median orbit simulations [ $\varepsilon = 23.5^{\circ}$ ; e = 0.0;  $\omega = 0^{\circ}$ ] of our study and De Vleeschouwer *et al.* (2014), as this is the simulation which is used for the ocean heat flux calibration in De Vleeschouwer *et al.* (2014). Finally, we present and discuss the requested maps showing the surface temperature differences between [ $\varepsilon = 23.5^{\circ}$ ; e = 0.069;  $\omega = 90^{\circ}$ ] and [ $\varepsilon = 23.5^{\circ}$ ; e = 0.069;  $\omega = 270^{\circ}$ ].

First, we investigate the response of continental snow to orbital forcing by discussing the differences between [ $\varepsilon = 23.5^{\circ}$ ; e = 0.069;  $\omega = 90^{\circ}$ ] and [ $\varepsilon = 23.5^{\circ}$ ; e = 0.069;  $\omega = 270^{\circ}$ ]. The upper panel of Figure 1 shows monthly differences between perihelion in December ( $\omega = 90^{\circ}$ ) and perihelion in June ( $\omega = 270^{\circ}$ ) in incoming solar radiation at Earth's surface, snow cover and temperature for the location shown in the lower panel. This figure can be compared to Figure 10 (c) in De Vleeschouwer *et al.* (2014).

The seasonal cycle of these variables in our model agrees very well with De Vleeschouwer *et al.* (2014): Solar radiation differences are negative from June to October, as the Earth receives less solar radiation during this season for perihelion in December. This leads to a slower melting of the Gondwanan snow cover for this orbital configuration which can be seen in the snow cover lines of De Vleeschouwer *et al.* (2014) and our study, showing a maximum in October. The snow albedo effect for perihelion in December is not as strong as observed in De Vleeschouwer *et al.* (2014) in our simulations, visible in the positive radiation difference for November and the earlier increase of temperature differences. However, we note that our location might slightly differ from the one De Vleeschouwer *et al.* (2014) use. We observe other regions further east in Gondwana with a longer persistence of the snow cover. Therefore, we conclude that the models agree very well with respect to the response of continental snow to orbital forcing and that this is not the root cause of the differences between our model results.

In the next step, we investigate the response of sea ice to orbital forcing by looking at the seasonal sea-ice fractions for different obliquities ([ $\varepsilon = 24.5$ ; e = 0.0;  $\omega = 0^{\circ}$ ],

Figure 1: Upper panel: Monthly means for incoming solar radiation, snow cover and surface air temperature at the location on Gondwana shown in the lower panel.

 $[\varepsilon = 23.5^{\circ}; e = 0.0; \omega = 0], [\varepsilon = 22.0^{\circ}; e = 0.0; \omega = 0]$ , Figure 2 to 4) and can compare them with sea-ice distributions in De Vleeschouwer *et al.* (2014) for the same orbital configurations (not shown here), finding significant differences.

As described in De Vleeschouwer *et al.* (2014), they find no sea ice for the obliquity maximum and attribute this to the thermal inertia of the oceans. This is a significant difference to the sea-ice distribution in our simulations (Figure 2): Comparing the Arctic sea-ice fraction of December for the obliquity maximum with the obliquity minimum, we also see the influence of the ocean's thermal inertia, as we have smaller fractions and a smaller extent of sea ice despite a smaller radiative forcing for the obliquity maximum for the Arctic in December. However, in contrast to De Vleeschouwer *et al.* (2014), sea ice does not disappear completely and follows the expected seasonal cycle: Sea ice starts to grow in December, the maximum arises in March due to a time lag caused by the thermal inertia of the oceans, and small fractions are left in June.

For the median orbit case, De Vleeschouwer et al. (2014) simulate no sea ice in the SH

and very small fractions for the Arctic. In our simulations, there is sea ice in the SH from June until September, Arctic sea ice reaches its maximum extent in March and shows a typical seasonal cycle (Figure 3).

For the obliquity minimum (Figure 4), we find comparable distributions and fractions for December for our simulation and De Vleeschouwer *et al.* (2014), but the seasonality differs significantly: For the Arctic, sea ice starts to grow in December in our model, reaches its maximum in March, decreases significantly until June and has vanished in September. In De Vleeschouwer *et al.* (2014), in contrast, sea ice starts to grow already in September–October–November (SON), reaches its maximum extent in December–January–February (DJF), recedes to a very small fraction in March–April–May (MAM) and is zero in June–July–August (JJA). In the Southern hemisphere (SH) we simulate sea ice from June until December with a maximum in September, whereas sea-ice fractions are very small in the SH in De Vleeschouwer *et al.* (2014) and increase only from MAM until JJA.

This analysis suggests that the ultimate cause of the differences between our results and De Vleeschouwer *et al.* (2014) results from the ocean/sea-ice components of the respective models. De Vleeschouwer *et al.* (2014) use a simpler mixed-layer ocean model with a heat-convergence term derived from the median orbit configuration and a thermodynamic sea-ice model which allows for free drift (Williams *et al.*, 2001). Our study, on the other hand, is based on an ocean general circulation model (Pacanowski & Griffies, 1999; Montoya *et al.*, 2005) and the two-dimensional, dynamic-thermodynamic sea-ice model by Fichefet & Maqueda (1997) which employs the elasto-viscous-plastic rheology of Hunke & Dukowicz (1997).

Moving on to the second part of our response, Figure 5 shows surface air temperature maps for the median orbit configuration [ $\varepsilon = 23.5^\circ$ ; e = 0.0;  $\omega = 0^\circ$ ]. Comparing this to Figure 2 in De Vleeschouwer *et al.* (2014), we see very good agreement. In Figure 6 we compare the ocean heat flux of our model with the one of De Vleeschouwer *et al.* (2014) (not shown) for the median orbit. Here again, we find a good agreement, with some differences on the coasts of Gondwana which might result from the differences in sea ice described above.

As requested in David De Vleeschouwer's comment, Figure 7 shows maps for the surface air temperature differences between perihelion in December [ $\varepsilon = 23.5^{\circ}$ ; e = 0.069;  $\omega = 90^{\circ}$ ] and perihelion in June [ $\varepsilon = 23.5^{\circ}$ ; e = 0.069;  $\omega = 270^{\circ}$ ], as in Figure 8 of De Vleeschouwer *et al.* (2014). The patterns over continental areas agree very well, suggesting again that the differences between the two studies are not due to land-albedo or atmospheric effects. In contrast to the agreement over land, however, the temperature differences over the Arctic ocean differ significantly: Arctic differences are negative for all seasons in De Vleeschouwer *et al.* (2014) and have a larger amplitude compared to our results. In our simulations, we have negative Arctic